# OpenReview forum: "On Uniform, Bayesian, and PAC-Bayesian Deep Ensembles"
_ICLR.cc/2025/Conference — ICLR 2025 Conference Withdrawn Submission_

### Official Review · Reviewer_bgX7 · 2024-11-04

**Soundness:** 3
**Presentation:** 2
**Contribution:** 2
**Rating:** 3
**Confidence:** 3

**Summary:**

This work focuses on the ensemble problem with uniform, Bayesian, and PAC-Bayesian variants. It summarizes and concludes that Bayes ensembles are not superior to uniform ensembles as it does not consider the cancellation of errors effect.  As for PAC-Bayesian ensembles, the authors leverage the tandem loss as the bound of the weighted majority vote. Then they propose to optimize the weights $\rho$ through the tandem loss.

**Strengths:**

1. This work focuses on an interesting problem of understanding the mechanism of ensembles.
2. It carries out extensive reviews of theoretical works regarding ensembles and provides a theoretical analysis of why Bayesian ensembles fail to select better weights compared with uniform ensembles.
3. The optimization of weights through the second-order bound and tandem loss is an interesting approach.

**Weaknesses:**

1. The derivation of eq. 3 is missing and the authors do not provide much insight into this equation. I suggest that the presentation and motivation can benefit from elaborating more on how this upper bound should be understood and what the implications could be.
2. The presentation does not properly highlight the contribution of this work. Section 3 summarizes many existing works and approaches while leaving its own contribution unclear. This section should be reorganized for a better presentation.
3. The proposed methodology is not properly described. For instance, the objective function is only discussed in one sentence in L185, without any formulation or definitions.
4. The main conclusion of this work is that the Bayesian Ensemble is no better than the simple deep ensemble approach. However, simple, uniform deep ensembles are already the conventional approach. Thus the contributions in this conclusion are pretty limited. As for the PAC-Bayesian ensemble, from the experimental results, it appears that the differences are very marginal. What are the benefits of optimizing this framework compared with the simple and conventional approach like deep ensembles?

[Minor]

1. $\hat{L}_{\mathcal{D}}$ in eq. 3 is not defined. The audience has to go through the reference to find out the definition. Also, the parenthesis and the bracket are reversed.
2. The colors of curves of the “-last” and “-all” variants should be more distinguishable.

**Questions:**

1. In eq. 3, is $\hat{L}_{\mathcal{D}}$ defined a the approximation of $L(h,h’)$ over $p(x,y)$ by sampling from $\mathcal{D}$? What is its expectation w.r.t. $\rho^2$? This shorthand notation should be defined.
2. The use of intermediate checkpoints has been extensively discussed in previous work such as snapshot ensembles. What are the contributions of this work regarding this point? Can the authors elaborate on their fourth contribution claimed in L96-99?
3. In L255-256, “we go from a distribution over the weights of a single neural network to a distribution over all parameters of the M networks and the ensemble weights." How is this implemented in practice? Can the authors give a clear formulation to demonstrate the difference?
4. Regarding the experiments, I have the following clarification questions:
    - In Figure 1: What are the differences between $\mathrm{AVG}_u$ and DE? From the definition in L117, $\mathrm{AVG}_u$ takes the uniform average, which is the vanilla approach of deep ensembles.
    - In Figure 1: Why are the uniform deep ensembles (DE) horizontal lines in Figure 2? It should also be affected by the number of ensemble members.
    - Why are the results of $\mathrm{AVG}_u$ omitted from Table 2 compared with Table 1?

---

### Official Review · Reviewer_hCWG · 2024-11-05

**Soundness:** 2
**Presentation:** 3
**Contribution:** 3
**Rating:** 5
**Confidence:** 3

**Summary:**

In this paper, the authors present their PAC-Bayesian Deep Ensembles which is a method for reweighing members in the posterior predictive of deep ensembles. The method in essence utilises a PAC-Bayesian bound to compute posterior predictive model weights of a deep ensemble through gradient optimisation on a hold-out dataset. In contrast to [1] which also investigates PAC-Bayesian bounds for posterior predictive weighting, the authors stress the use of second order PAC-Bayesian analysis. The authors also discuss why deep ensembles often outperform single mode Bayesian Neural Networks (referred to as Bayes ensembles in the paper), and motivate PAC-Bayesian bounds from this. The authors evaluate the methods on a number of datasets and compare to other methods.


[1]: Diversity and Generalization in Neural Network Ensembles, Ortega et. al, 2021

**Strengths:**

* The paper is well written.
* The authors have generally performed a thorough literature review.
* The method and general area of research is of high interest in the community (methods for aiding model generalisation, robustness and uncertainty quantification).

**Weaknesses:**

I will outline the weaknesses here and elaborate on them in the questions.

* Lack of novelty. This is the main reason for the low score.
* Lacking experimental evaluation metrics. Given the lack of novelty, I would strongly argue that experimental evaluation should be much larger for this work to be accepted.

**Questions:**

I will here outline my reasons for the listed weaknesses.

* With regards to lack of novelty, I here specifically outline why the listed main contributions in my opinion are not novel:
    1. As far as I am aware, the concept of reweighing ensemble members post model training is not new. If I have understood the proposed method correctly, then this is what is commonly referred to as _stacking_ in ensembles. See e.g. [1].
    2. "We discuss the conceptual differences... ". The discussion on why deep ensembles outperform single mode bayesian neural networks (referred to as Bayes ensembles) is commonly known, understood and researched within the Bayesian neural network community. See e.g. [2]. This also relates to the third listed main contribution which has previously been shown and is widely understood.
* In my opinion some key metrics are lacking. Currently the authors only report accuracy, but I think negative log-likelihood or calibration metrics (ECE) are equally important and of interest to the Bayesian community. Currently there are no metrics showcasing uncertainty quantification of the proposed methods which is of high interest in these types of works. Could the authors possible include some of these metrics?
* Could the authors discuss why they write "As outlined above, general ensemble methods and Bayesian neural networks have different motivations" in line 195-196? In my opinion they are motivated for the same reason: improved generalisation, robustness and uncertainty quantification. One simply is multimodal (ensembles) whilst the other (typically) only covers a single mode (BNNs).
* I strongly disagree with the sentence in lines 480-481 "Bayesian model averaging (BMA) is not meant to maximize predictive performance but to get a better idea about the maximum likelihood estimate of the parameters of a single model.". To me, the exact point of BMA is to improve model uncertainty quantification and predictive performance. Could the authors elaborate on this?

Minor comments/questions:
1. In line 133 the authors write "It is in general difficult to sample from $p(w|D)$, which can .. ", but I suspect the authors meant to say that it is difficult to approximate $p(w|D)$?
2. The setup in the results section is a little bit confusing to me when reading the paper. I would suggest that the authors perhaps first present experimental setup, and then discuss results (but this is an opinion).
3. The naming convention of "Bayes ensembles" is confusing to me. I have not seen this terminology before and to me it indicates an ensemble of Bayesian neural networks. Are the authors in fact just referring to Bayesian neural networks?

[1]: Stacking for Non-mixing Bayesian Computations: The Curse and Blessing of Multimodal Posteriors, Yao et. al, 2018.

[2]: Bayesian Deep Learning and a Probabilistic Perspective of Generalization, Wilson and Izmailov, 2020.

---

### Official Review · Reviewer_JLRF · 2024-11-07

**Soundness:** 3
**Presentation:** 3
**Contribution:** 1
**Rating:** 3
**Confidence:** 3

**Summary:**

The paper examines the effectiveness of different ensemble techniques—uniform, Bayesian, and PAC-Bayesian—specifically applied to deep neural networks. The authors contend that Bayesian ensembles, which typically average models based on posterior distributions, are suboptimal for maximizing generalization performance due to limitations in supporting the "cancellation of errors" effect. This issue arises because Bayesian ensemble members tend to converge to similar predictions rather than compensating for each other's errors. Instead, the authors propose a PAC-Bayesian approach that optimizes ensemble weights by minimizing a tandem loss over the validation set. This method accounts for model correlations and aims to increase robustness by improving generalization, even when ensemble members are generated from similar training processes. The empirical results on several benchmark datasets show that PAC-Bayesian ensembles with optimized weighting  using the validation set outperform Bayesian ensembles.

**Strengths:**

- Studies deep ensembles which is an important and interesting problem
- Brings together different types of ensembles studied in the literature together in a unified framework.
- Includes empirical results supporting the theoretical claims

**Weaknesses:**

* Using the Bernstein-von Mises theorem theorem to show that a Bayes ensemble corresponds to a single model and hence cannot capture the cancellation of errors effect is interesting. But as mentioned by the paper, this has already been shown by the Masegosa paper already.
* Not clear what this paper is adding over the Masegosa paper
* This work claims that “Our results on four datasets show that complex Bayesian approximate inference methods can often be surpassed by more efficient simple deep ensembles.” but that has already been known empirically. This is not a novel observation.
* “• We show that the optimization of a PAC-Bayesian generalization bound using the tandem loss can improve the predictive performance of deep ensembles and provides non-vacuous generalization guarantees.”. Not clear what are they adding over the Masegosa paper.
* ”We demonstrate that the inclusion of intermediate checkpoints from the same training run in a neural network ensemble can increase its predictive performance, especially when the PAC-Bayesian bound optimization weights their contributions.” This has also been shown by snapshot ensembles as already mentioned by this paper. It is not clear what is the novel contribution in this regard also.
*  The main contribution of the paper seems to be using the validation set to optimize the weights of the ensemble models which does not seem to be too novel.
* Overall, in the end the authors are able to match the performance of Pac-bayesian ensembles to that of deep ensembles after using the hold-out data. There does not seem to be any gains in using their approach.

**Questions:**

* Using the Bernstein-von Mises theorem, the authors require some assumptions mentioned in line 219. Can the the authors clarify if these assumptions hold for the deep networks class?

---

### Official Review · Reviewer_bVKq · 2024-11-12

**Soundness:** 2
**Presentation:** 2
**Contribution:** 2
**Rating:** 3
**Confidence:** 3

**Summary:**

This paper rethinks three methods (i.e., uniform, Bayesian, and PAC-Bayesian) for deep ensembles. The authors argue that neither the sampling nor the weighting in a Bayes ensemble is well-suited for increasing generalization performance. In comparison, a weighted average of models obtained by minimizing a PAC-Bayesian generalization bound can improve the performance. Empirical results are conducted to support these hypotheses.

**Strengths:**

1. Overall, this paper is well-written and the main hypotheses or claims are clearly presented.
2. Experimental results support these hypotheses.

**Weaknesses:**

1. Theoretical results are lacking to support these hypotheses or claims.
2. The authors have not proposed a new method or theory, resulting in little contribution.

**Questions:**

None

---

### Note · Authors · 2024-12-19

**Comment:**

We would like to thank Reviewer bgX7 for engaging in a discussion.

**Withdrawal Confirmation:**

I have read and agree with the venue's withdrawal policy on behalf of myself and my co-authors.